# Predicting child development and school readiness, at age 5, for Aboriginal and non-Aboriginal children in Australia's Northern Territory

**Abel Fekadu Dadi**[1]*, **Vincent He**[1], **Georgina Nutton**[2], **Jiunn-Yih Su**[1], **Steven Guthridge**[1]

**1** Menzies School of Health Research, Charles Darwin University, Darwin, Northern Territory, Australia, **2** College of Indigenous Futures, Education and the Arts, Charles Darwin University, Darwin Northern Territory, Australia

\* abel.dadi@menzies.edu.au

**Data Availability Statement:** The study datasets contain sensitive personal information and are held on a secure cloud-based server with restricted access. Access requires the approval of the ethics

## Abstract

### Background

Positive early development is critical in shaping children's lifelong health and wellbeing. Identifying children at risk of poor development is important in targeting early interventions to children and families most in need of support. We aimed to develop a predictive model that could inform early support for vulnerable children.

### Methods

We analysed linked administrative records for a birth cohort of 2,380 Northern Territory children (including 1,222 Aboriginal children) who were in their first year of school in 2015 and had a completed record from the Australian Early Development Census (AEDC). The AEDC measures early child development (school readiness) across five domains of development. We fitted prediction models, for AEDC weighted summary scores, using a Partial Least Square Structural Equation Model (PLS-SEM) considering four groups of factors–pre-pregnancy, pregnancy, known at birth, and child-related factors. We first assessed the models' internal validity and then the out-of-sample predictive power (external validity) using the PLS$_{predict}$ procedure.

### Result

We identified separate predictive models, with a good fit, for Aboriginal and non-Aboriginal children. For Aboriginal children, a significant pre-pregnancy predictor of better outcomes was higher socioeconomic status (direct, $\beta = 0.22$ and indirect, $\beta = 0.16$). Pregnancy factors (gestational diabetes and maternal smoking (indirect, $\beta = -0.09$) and child-related factors (English as a second language and not attending preschool (direct, $\beta = -0.28$) predicted poorer outcomes. Further, pregnancy and child-related factors partially mediated the effects of pre-pregnancy factors; and child-related factors fully mediated the effects of pregnancy

committee and data custodians. For applications for data access, please contact the Menzies Data-linkage Program Leader at steve.guthridge@menzies.edu.au, or Mike Robbins, the Menzies Research and Data Systems Manager at datamanagement@menzies.edu.au.

**Funding:** The study was supported by a grant from the Northern Territory Government through the Child and Youth Development Research Partnership. The funding body was not involved in the study design, data analysis, interpretation of data, or preparation and publication of the research paper. There was no additional external funding received for this study.

**Competing interests:** The authors have declared that no competing interest exist.

**Abbreviations:** NT, Northern Territory; AEDC, Australian Early Development Census; PLS-SEM, Partial Least Square Structural Equation Model; AEDI, Australian Early Development Index; CYDRP, Child and Youth Development Research Partnership; IRSD, Relative Socio-Economic Deprivation; VIF, variance inflation factor; SMSA, Standardized Mean Square Residual; NFI, Normal Fit Index; RMSE, Root Mean Squared Error; CCA, complete case analysis; MICE, multiple imputation by chained equation.

factors on AEDC weighted scores. For non-Aboriginal children, pre-pregnancy factors (increasing maternal age, socioeconomic status, parity, and occupation of the primary carer) directly predicted better outcomes ($\beta = 0.29$). A technical observation was that variance in AEDC weighted scores was not equally captured across all five AEDC domains; for Aboriginal children results were based on only three domains (emotional maturity; social competence, and language and cognitive skills (school-based)) and for non-Aboriginal children, on a single domain (language and cognitive skills (school-based)).

## Conclusion

The models give insight into the interplay of multiple factors at different stages of a child's development and inform service and policy responses. Recruiting children and their families for early support programs should consider both the direct effects of the predictors and their interactions. The content and application of the AEDC measurement need to be strengthened to ensure all domains of a child's development are captured equally.

## Background

Early childhood is a critical period that lays the foundations for a child's development, long-term health, and lifelong learning [1–3]. It is also the period in which inequalities experienced by disadvantaged populations are established and potentially persist into adolescence and adulthood. As such, early childhood is widely recognised as one of the best opportunities to overcome inequalities through investment in early interventions that provide additional support to disadvantaged children and families.

The etiology of delay in children's early developmental outcome is multifactorial [4], and includes genetic and environmental factors, and their interaction [5,6]. Although recent genome-wide association studies have suggested a significant role of genetic factors in child development [7,8], the direct and indirect effects of the environment are paramount [7,9]. Environmental factors associated with early development include poverty or family income [10–12], neighborhood economic disadvantage [13–15], maternal employment [16], parental nurturing [17], maternal age [18], sex of the newborn, and preschool attendance and absenteeism [19]. In utero influences from maternal nutrition and behavior (smoking, alcohol and other drug consumption) [10,20–25], newborn nutrition [26–30], adverse birth outcomes (preterm and low birth weight) [31–38], lack of optimal breastfeeding [39], and psychosocial risk factors (maternal mental health, neighborhood cohesion, childhood maltreatment, and parental offensive behavior) [40–45] are additional factors that contribute to poor child development.

Recognising the importance of routinely collected data to inform efficient service delivery and address systemic inequalities in early childhood, in 2009, Australia became the first country in the world to conduct a national census of childhood development for all children starting school [46–48]. This census was initially known as the Australian Early Development Index (AEDI) and from 2014 has been known as the Australian Early Development Census (AEDC) [46–48]. The AEDC measures early childhood development at school entry (usually age 5) across five distinct domains: physical health and wellbeing, social competence, emotional maturity, language and cognitive skills (school-based), and communication skills and general knowledge [47,48].

Australian Aboriginal and Torres Strait Islander people (hereafter respectfully referred to as Aboriginal people) experience significant educational, health and social inequalities because of a history of colonisation and adverse social policies across generations. Starting from the pre-natal period, Aboriginal children are much more likely to face multiple social and health disad-vantages than other Australian children. This inequality continues in the first year of full-time schooling, at around age 5. Although the proportion of Aboriginal children identified as devel-opmentally vulnerable has decreased from 47.4% in 2009 to 41.3% in 2018 (defined as develop-mentally vulnerable on one or more AEDC domains), this proportion remains almost twice that of non-Aboriginal children (20.4% in 2018) [48]. The disparity experienced by Aboriginal children persists after controlling for non-English speaking background, jurisdiction and socioeconomic condition of the child's community of residence, demonstrated by Brinkman et al after analysing the 2009 AEDC data from Australian states and territories [49]. Analysis of AEDC data has also highlighted one potential area to improve educational outcomes and reduce educational inequality—preschool participation [48]. Nonetheless by 2021, more than ten years after the Australian governments' 'Closing the Gap' agreement to overcome. Indige-nous disadvantage only 41% of Aboriginal children attended 15 hours of preschool a week [50]. A new National Agreement on Closing the Gap was approved in 2020 and included two targets relating to early childhood development—(a) to increase the proportion of Aboriginal children enrolled in early childhood education to 95% by 2025; and, (b) to increase the propor-tion of Aboriginal children assessed as developmentally on track in all five AEDC domains to 55% by 2031 [51].

The Northern Territory (NT) is a large, remote and sparsely populated area of northern and central Australia. The NT has distinct demography compared to other states and territo-ries in Australia, with: the smallest total population (approximately 247,300) [52]; a much greater proportion of Aboriginal children (39.7% compared with 4.8% for Australia as a whole); enrolling in remote or very remote schools (47.8% c/w 3.0%); and, with language back-ground other than English at school entry (39.4% c/w 18.0%) [48,53]. The gap between Aboriginal and non-Aboriginal children in early developmental outcomes is greater in the Northern Territory (NT) than in other states and territories of Australia. The first study of NT children's early developmental outcomes, based on AEDC, found a higher proportion of Aboriginal children identified as developmentally vulnerable on one or more AEDC domain (s) than non-Aboriginal children (68.3% c/w 23.2%), with half of this difference explained by potentially modifiable early health and sociodemographic factors [54]. A more recent study found that once these modifiable factors were adjusted, in multivariable analysis, Indigenous status was no longer associated with developmental vulnerability and that the main influences associated with children's developmental outcome were early life health and sociodemographic factors [55]. The findings from these previous studies suggest potential levers to 'close the gap' in educational outcomes in Australia [51], with the need to recognise NT's distinct characteristics.

In the NT there is a substantial overlap between Aboriginal children and children with a non-English speaking background. Brinkman et al found that less than 1% of all Australian children (excluding the NT) taking AEDC assessment have both Aboriginal heritage and non-English speaking background [56]. It is not clear how the higher proportion of students with non-English speaking background affects the interpretation of the AEDC results, given that two of the five AEDC domains are related to language (language and cognitive skills (school based), and communication skills and general knowledge) [49]. Brinkman et al had earlier noted that students are often required to demonstrate competencies in English *"data for some children with language backgrounds other than English will continue to reflect developing English proficiency, rather than developmental deficiency of lack of capacity"* [56]. There have also been

concerns about AEDC's reliance on "Standard Australian English (SAE) competence as the basis of language and cognition assessment" [57]. Brinkman et al noted that AEDC is based on "*the empirical research conducted in western contexts*" to support "*children to take advantage of the school learning environment*", and that "*the embedded cultural constructs do not necessarily reflect the early childhood experiences of the entire population*" [49]. Similarly, Bell et al utilised AEDC to conceptualise 'school readiness', acknowledging that this conceptualisation "*comes from a predominantly Western perspective, in that abilities included in the school readiness measure are those that are considered important for achieving success in a Western education context. As such, the measurement and interpretation of school readiness is conducted within the context of Western social and cultural norms regarding what is important for school adaption*" [58]. Recognising the underlying principles of AEDC to support school-based learning and that AEDC measures 'school readiness' for a Western style education system this study adopted a similar stand to Bell (2017) and uses the compound term child development (school readiness) to describe AEDC outcomes reported in this study [58].

While previous NT studies have identified risk factors of early developmental outcomes separately for both Aboriginal and non-Aboriginal children [54,55], there remains a need for a tool that predicts those children most at risk of poor school outcomes to inform a nuanced approach to improve schooling outcomes. Studies conducted so far are explanatory [18,24,36,54,59,60], and have not accounted for the inherent theoretical philosophy imbedded in life span perspective of human development [61] nor the latent nature of the AEDC construct [10,24]. These analytic approaches do not reflect the statistical and theoretical philosophies underlying childhood development, while the dichotomization of the AEDC measurement may introduce measurement error [62]. Expanding on previous studies, this predictive study uses a machine learning approach, Partial Least Squares Structural Equation Modelling (PLS-SEM), to identify the predictors of and pathways to school readiness at school entry.

## Methods

### Design and population

This was a retrospective cohort study of children, using linked, individual-level records from NT administrative datasets. We included all children who were in their first year of schooling in 2015 [63] and for whom all AEDC domains had been completed. Children with previously identified special needs were not included within AEDC domain indicator/categories.

### Data sources and linkage

Datasets used in this study were sourced from an extensive repository of linked administrative datasets maintained by the Child and Youth Development Research Partnership (CYDRP), a collaboration between Menzies School of Health Research and seven NT Government agencies. The first stage of data linkage was completed by SA NT DataLink using probabilistic linkage with clerical review for uncertain matches [64]. After completion of the first stage linkage unique person specific linkage keys were provided to data custodians who prepared research datasets containing the linkage keys and approved data variables but with removal of all identifiers. Each research dataset was then provided by the respective data custodian to the investigators who were able to use the linkage keys to link records for individuals across multiple datasets and prepare data for analysis. The detailed data linkage process has been described elsewhere [54]. For the analysis, we combined records from the AEDC dataset with the following datasets: child protection service (CPS) contacts, education, hospital admissions and the NT Perinatal Data Register.

## Outcome variable and measurement

The outcome variable for this study was early child development (school readiness) measured using the summary scale score of AEDC. The AEDC was collected on children in their first year of school enrolment (usually aged between five and six years) by classroom teachers. The Australian version of the Early Development Instrument (AvEDI) used for the AEDC contains 96 items across the five domains: physical health and wellbeing, social competence, emotional maturity, language and cognitive skills (school-based), and communication skills and general knowledge [65]. For each of the five AEDC domains, children received a score between 0 and 10 where 0 indicated most developmentally vulnerable [47]. Validation studies of the Early Development Instrument (EDI) have been conducted in Canada, Australia, and USA and demonstrated fair to good validity [66].

## Covariates and data sources

The potential covariates accessed from the AEDC, perinatal, child protection, and hospital admission datasets were linked to form a dataset for analysis. We adopted the inherent life span perspective of human development for conceptualizing the potential causal mechanisms underlying child development (school readiness) at age five [67] (Fig 1).

The following covariates were organized in line with this framework:

i. Pre-pregnancy covariates included socio-economic disadvantage measured using the Index of Relative Socio-Economic Deprivation (IRSD) [68], Indigenous status, relative remoteness (using the Accessibility and Remoteness Index of Australia) [69], mother's education,

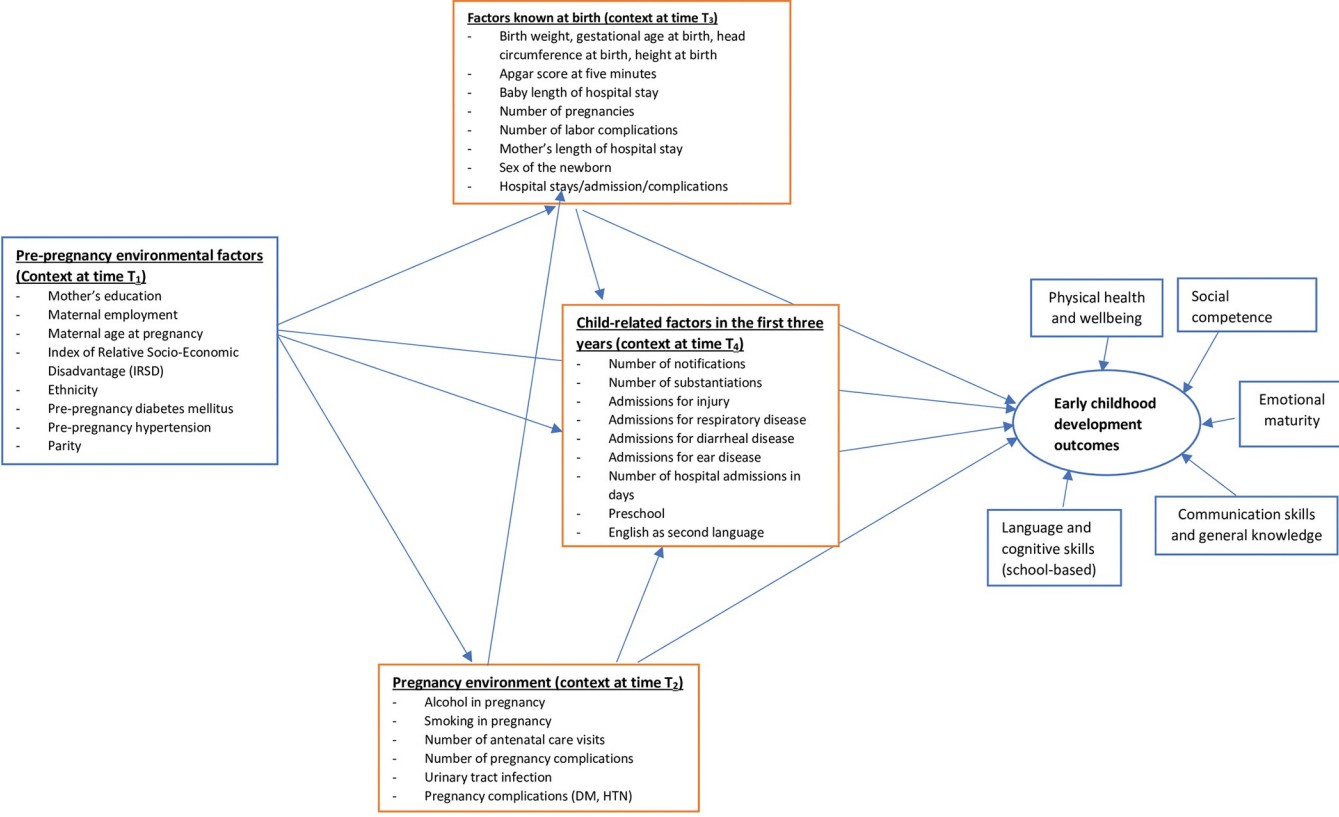

**Fig 1. Conceptual framework for early childhood development.**

occupation, maternal age at pregnancy, parity, and maternal pre-existing medical conditions (diabetes mellitus, hypertension).

ii. Pregnancy covariates included smoking, alcohol consumption, number of antenatal care visits, urinary tract infection, obstetric morbidities (hypertension, gestational diabetes, preeclampsia, cardiac disease, renal disease).

iii. Factors known at birth included the gender of the child, fetal distress, Apgar score at five minutes, gestational age, birth weight in kilograms, birth length in centimetres, head circumference at birth in centimetres, resuscitation required, twin birth, type of delivery, mother's and child's length of hospital stay before and after delivery.

iv. Child-related covariates included preschool attendance and English as a second language. Other covariates (number of child protection notifications, number of child protection substantiations, older siblings with child protection service contact, hospital admission for gastroenteritis, hospital admission for injury and hospital admission with an acute lower respiratory infection) were restricted to the first three years to maintain the temporal relationships with the outcome. The interaction and mediation effect of these factors on the AEDC outcome measures were also tested [70].

## Data analysis

We checked the linked data for completeness and quality prior to fitting regression models. We conducted correlation, chi-square, and t-tests to check for crude relationships between the outcome and predictor variables. As the outcome to be predicted is a latent variable, fitting prediction models went through two steps. In the first step we identified potential predictor variables using regression analysis. In this step the total AEDC summary scale scores (the sum of the scale scores for all five domains) were used as a continuous outcome variable. We checked the distribution of the summary scale scores in Aboriginal and non-Aboriginal children. We found the distribution of scale scores to be approximately normal in Aboriginal children but significantly skewed in their non-Aboriginal peers. Further, to account for unobserved clustering effects for children from the same teacher (who administered the AEDC assessment) and school (where the teachers work), we fitted a multilevel model (Appendix I in S1 File). For Aboriginal children, we fitted a mixed linear model using total AEDC summary scale scores. For non-Aboriginal children we fitted a multilevel logistic regression with robust standard error [71] using results for each of the five domains as well as the summary measure 'being developmentally vulnerable in two or more domains' (DV2) [72]. We fitted separate prediction models for Aboriginal and non-Aboriginal children for the following reasons: (i) there were differences in the number and type of predictors we identified in the regression analysis (Appendix II in S1 File); (ii) there were differences in the distributions of the AEDC measurement; and (iii) the AEDC domains were not equally captured by Aboriginal and non-Aboriginal children [71].

In the second step we fitted a predictive model using the identified predictors (with $p< = 0.1$) from regression analysis following the life course framework (Fig 1). As AEDC scores are a latent construct representing child development (school readiness) and the intention of this analysis was prediction, we fitted the Partial Least Squares Structural Equation Model (PLS-SEM) [73]. This model has been recommended for use in developmental studies [74]. The PLS-SEM is a composite-based model in which the latent variables are measured as a weighted composites of their indicators to maximize their explained variance [75]. The composite scores were computed as a weighted sum of the set of indicators, and the variance of the

component was equivalent to one. In the final component score estimates, the path coefficients and loadings were estimated by ordinary least squares and the standard errors were estimated via the bootstrap method [76].

The five domains of AEDC were measured formatively and we fitted a formative measurement model. The assumption behind formative measurement is that the AEDC is formed by an equal non-replaceable representation of the five domains. We set the PLS-SEM algorithm for path weighting to 300 maximum iterations with a stopping criteria of $10^{-7}$ [77]. We assessed the measurement model for collinearity of the indicators (variance inflation factor (VIF) > 5 is not acceptable), and significance and relevance of indicators outer weights (recommended to be greater than 0.5 and significant) [78]. After having checked for satisfactory fit of the measurement model, we assessed the in-sample (in the study sample population) fit of the structural model using a coefficient of determination ($R^2$), significance and relevance of structural path coefficients, Standardized Mean Square Residual (SMSA < 0.08), and Normal Fit Index (NFI > 0.9). We then assessed the model's in-sample predictive relevance or predictive accuracy ($Q^2$) by running a blindfolding-based cross-validated redundancy with omission distance of seven (should be above zero) [78,79].

We finally assessed the out-of-sample (in the extrapolated sample population) predictive power of the model using the PLS$_{predict}$ procedure. The PLS$_{predict}$ procedure generates a hold-out sample and use this sample to test the out-of-sample predictive power of the model [80]. The PLS$_{predict}$ uses a concept of k-fold cross-validation. This means, the procedure splits the dataset into k equal sized parts and uses the combined k-1 datasets as a training sample to develop prediction model and the remaining subset as a holdout sample for validating the model [81]. We set the PLS$_{predict}$ algorithm to do a 10-fold cross validation with 10 times repetition [80]. The out-of-sample predictive relevance of the predictive model ($Q^2_{predict}$), Mean Absolute Error (MAE), and the Root Mean Squared Error (RMSE) were used to evaluate model out-of-sample predictive power. These prediction statistics and the model selection indices (AIC & BIC) were used for comparing the in-sample and out-of-sample predictive ability of different competing predictive models [81]. We used prediction errors distribution obtained from the PLS-SEM and linear model to choose between the two preferred prediction statistics (MAE or RMSE). We used a guideline developed by Shmueli et al to judge the out-of-sample prediction power of the models [81]. Lastly, we selected and interpreted a non-recursive (simple) model with better predictive power. All analyses were conducted in Stata 16 and SmartPLS version 3.2 [82].

## Missing data handling

Our preliminary data checking found six variables contain missing data: maternal education (30.4% of records), mothers' occupation (26.5%), maternal smoking in pregnancy (26.2%), maternal alcohol consumption in pregnancy (9.4%), newborn head circumference (19.4%), and child preschool education (2.7%). Aboriginal children were more likely to have missing observations than non-Aboriginal children in all six variables (Table 1). The missing mechanisms is potentially missing not at random (MNAR) as the reason for missingness was explained by other missing covariates and the missing values themselves might be more likely to correlate with the missing values (for example a woman with low level of education is more likely to escape the question because of social desirability issues). We also checked that the missingness was not dependent on the outcome (AEDC) after adjusting for the other covariates. We checked this by first creating a missing indicator variable for each variable with missingness (yes = 1, no) and then fitting regression to model the missingness accounting for other covariates. In this case, methodological studies have reported that multiple imputation (MI)

**Table 1. Pre-pregnancy and pregnancy characteristics and their univariate association with AEDC scale score (N = 2380).**

| Variables | Aboriginal children | P-value | Non-Aboriginal children | P-value |
|---|---|---|---|---|
| | N (%) | | N (%) | |
| *Pre-pregnancy covariates* | | | | |
| Indigenous status | 1,158 (48.7) | | 1,222 (51.3) | |
| Remoteness | | <0.001 | | 0.993 |
| Outer regional | 248 (21.4) | | 898 (73.5) | |
| Remote | 272 (23.5) | | 268 (21.9) | |
| Very remote | 638 (55.1) | | 56 (4.6) | |
| **IRSD (mean, ±SD)** | 2.45 (3.13) | <0.005 | 6.16 (2.65) | <0.005 |
| Mother's education | | <0.001 | | 0.001 |
| Non-school qualification | 429 (62.7) | | 238 (24.5) | |
| Certificate level 1–4 | 176 (25.7) | | 292 (30.0) | |
| Advanced diploma/diploma | 44 (6.4) | | 141 (14.5) | |
| Bachelor's degree or above | 35 (5.1) | | 302 (31.0) | |
| Missing | 474 (40.9) | | 249 (20.4) | |
| Mother's occupation | | <0.001 | | 0.024 |
| Unemployed (Group 5) | 489 (42.2) | | 260 (21.3) | |
| Group 4 | 86 (7.4) | | 84 (6.9) | |
| Group 3 | 90 (7.8) | | 261 (21.4) | |
| Group 2 | 29 (2.5) | | 163 (13.3) | |
| Group 1 | 52 (4.5) | | 235 (19.2) | |
| Missing | 412 (35.6) | | 219 (17.9) | |
| Mother's age at pregnancy (mean, ±SD) | 25(6.3) | 0.910 | 30.2(5.8) | 0.085 |
| Parity (mean, ±SD) | 3 (1, 4) | | 2 (1, 3) | |
| Maternal complication: Pre-existing diabetes | | 0.150 | | 0.380 |
| No | 1,135 (98.0) | | 1,212 (99.2) | |
| Yes | 23 (2.0) | | 10 (0.8) | |
| Maternal complications: Pre-existing hypertension | | 0.979 | | 0.711 |
| No | 1,126 (97.2) | | 1,218 (99.7) | |
| Yes | 32 (2.8) | | 4 (0.3) | |
| *Pregnancy covariates* | | | | |
| Mother smoked during pregnancy | | 0.023 | | 0.005 |
| No | 360 (31.1) | | 833 (68.2) | |
| Yes | 434 (37.5) | | 129 (10.5) | |
| Missing | 364 (31.4) | | 260 (21.3) | |
| Mother consumed alcohol in pregnancy | | 0.006 | | 0.776 |
| No | 862 (74.4) | | 1,124 (92.0) | |
| Yes | 133 (11.5) | | 37 (3.0) | |
| Missing | 163 (14.1) | | 61 (5.0) | |
| Number of antenatal visits (mean, ±SD) | 8.6 (4.7) | 0.337 | 9.7 (3.2) | 0.540 |
| Urinary tract infection during pregnancy | | 0.312 | | 0.287 |
| No | 1,122 (96.9) | | 1,209 (98.9) | |
| Yes | 36 (3.1) | | 13 (1.1) | |
| History of obstetric complications | | 0.024 | | 0.020 |
| No | 700 (60.4) | | 981 (80.3) | |
| Yes | 458 (39.6) | | 241 (19.7) | |
| Gestational diabetes | | 0.041 | | 0.376 |
| No | 1,069 (92.3) | | 1,160 (94.9) | |

(*Continued*)

**Table 1.** (Continued)

| Variables | Aboriginal children | P-value | Non-Aboriginal children | P-value |
|---|---|---|---|---|
| | N (%) | | N (%) | |
| Yes | 89 (7.7) | | 62 (5.1) | |
| Pre-eclampsia | | 0.030 | | 0.303 |
| No | 1,114 (96.2) | | 1,196 (97.9) | |
| Yes | 44 (3.8) | | 26 (2.1) | |

Note: p-value for association between risk factors and AEDC scale score for Aboriginal and for non-Aboriginal children.

tends to be biased away from the null and complete case analysis (CCA) is a better approach [83,84]. We therefore used potential predictors (predictors with p-value up to 0.1) identified from CCA in the regression analysis to fit the predictive models. As a sensitivity analysis, we fitted two models that applied the two methods of handling missing and compared the results: (i) missing indicator, considering missingness as a separate category, and (ii) MI by chained equations (MICE) and found them to retain different covariates in the final models to those identified with CCA (Appendix II in S1 File).

### Ethics approval

The study was approved by the Human Research Ethics Committee of the NT Department of Health and the Menzies School of Health Research (HREC-2016-2708). A waiver of consent was approved for this study which is based on de-identified population level data. The study was also approved by the First Nations Advisory Group for the CYDRP Program. The group includes independent community members.

## Results

### Pre-pregnancy and pregnancy characteristics of the study participants

There were 2380 children in the study cohort (represented around 67% of the total population). The study participant characteristics and their univariate association are displayed in Table 1. Slightly more than half (51.3%) of the children in this study were non-Aboriginal children. Most non-Aboriginal children (73.5%) were from an outer regional area (Darwin and surrounds). About 31.0% of mothers of non-Aboriginal children had a Bachelor's degree or higher qualification while more than half of the mothers of Aboriginal children (62.7%) had no formal school qualification. Maternal education and occupation had significant association with AEDC in both Aboriginal and non-Aboriginal children. A higher proportion of mothers of Aboriginal children had a record of smoking (37.5% Vs 10.5%) and alcohol consumption (11.5% Vs 3.0%) during pregnancy compared to mothers of non-Aboriginal children. There was evidence that both smoking during pregnancy and any record of an obstetric complication were associated with lower AEDC scores for both Aboriginal and non-Aboriginal children.

### Birth and child-related characteristics of the study participants

Birth and child-related study participants characteristics and univariate association are presented in Table 2. Gender of the newborn and birth length are the two covariates known at birth for which there was evidence of an association with AEDC outcomes for both Aboriginal and non-Aboriginal children. More than two-thirds of Aboriginal (70.6%) and three-quarters of non-Aboriginal (88.9%) children attended preschool, with preschool attendance associated

**Table 2. Birth and child-related characteristics of study participants and their univariate association with AEDC scale score (N = 2380).**

| Variables | Aboriginal children | P-value | Non-Aboriginal children | P-value |
|---|---|---|---|---|
| | N (%) | | N (%) | |
| *Known at birth covariates* | | | | |
| Gender of the child | | <0.001 | | <0.001 |
| Male | 600 (51.8) | | 622 (50.9) | |
| Female | 558(48.2) | | 600 (49.1) | |
| Postpartum hemorrhage | | 0.052 | | 0.116 |
| No | 947 (81.8) | | 1,069 (87.5) | |
| Yes | 211 (18.2) | | 153 (12.5) | |
| Apgar score at five minutes (median, IQR) | 9 (9,9) | 0.351 | 9 (9,9) | 0.075 |
| Gestational age (mean, ±SD) | 38.6 (2.1) | 0.059 | 38.9 (1.9) | <0.001 |
| Birth weight (mean, ±SD) | 3.2 (0.6) | 0.112 | 3.4 (0.6) | 0.003 |
| Birth length in cm (mean, ±SD) | 46.8 (12.1) | 0.030 | 32.6 (24.5) | 0.021 |
| Head circumference at birth (mean, ±SD) | 32.7 (8.7) | 0.491 | 33.3 (10.4) | 0.647 |
| Resuscitation required | | 0.968 | | 0.818 |
| No | 631 (54.5) | | 598 (48.9) | |
| Yes | 527 (46.5) | | 624 (52.1) | |
| Twin birth | | 0.003 | | 0.818 |
| No | 1,125 (97.1) | | 1,178 (96.4) | |
| Yes | 33 (2.9) | | 44 (3.7) | |
| Type of delivery | | 0.045 | | 0.929 |
| Normal | 973 (84.0) | | 1,025 (83.9) | |
| Emergency caesarean section | 185 (16.0) | | 197 (6.1) | |
| Mother's hospital stays before and after delivery (median, IQR) in days | 4 (2, 6) | <0.001 | 3 (2, 5) | 0.635 |
| *Child-related covariates* | | | | |
| Child preschool attendance | | <0.001 | | 0.110 |
| No | 123 (10.6) | | 51 (4.2) | |
| Yes | 818 (70.6) | | 1,086 (88.9) | |
| English as a second language | | <0.001 | | 0.016 |
| Yes | 806 (69.6) | | 138 (11.3) | |
| No | 352 (30.4) | | 1,084 (88.7) | |
| Number of child protection notifications in the first three years (minimum, maximum) | (0, 7) | 0.002 | (0, 9) | 0.001 |
| Number of substantiations in the first three years (minimum, maximum) | (0, 3) | 0.029 | (0, 4) | 0.146 |
| Older sibling with child protection record | | 0.003 | | 0.122 |
| No | 873 (75.4) | | 1,185 (97.0) | |
| Yes | 285 (24.6) | | 37 (3.0) | |
| Admission for gastroenteritis | | <0.001 | | 0.002 |
| No | 889 (76.8) | | 1,170 (95.7) | |
| Yes | 269 (23.2) | | 52 (4.3) | |
| Admission for nutritional deficiency | | <0.001 | | 0.155 |
| No | 935 (80.7) | | 1,213 (99.3) | |
| Yes | 223 (19.3) | | 9 (0.7) | |
| Admission for acute lower respiratory infection | | <0.001 | | 0.053 |
| No | 805 (69.5) | | 1,155 (94.5) | |
| Yes | 353 (30.5) | | 67 (5.5) | |
| Child developmental outcome | | | | |
| Total AEDC scale scores (median, IQR) | 34.3(26.2, 42.0) | <0.001[#] | 45.2 (39.6, 48.4) | |

(*Continued*)

**Table 2.** (Continued)

| Variables | Aboriginal children | P-value | Non-Aboriginal children | P-value |
|---|---|---|---|---|
|  | N (%) |  | N (%) |  |
| Total AEDC scale scores (mean, ±SD) | 33.7(9.9) | <0.001[#] | 43.2(6.6) |  |

*Notes: p-value for association between risk factors and AEDC scale score for Aboriginal children and for non-Aboriginal children. [#] p-value for comparison of scores between Aboriginal children and non-Aboriginal children.*

with the outcome for Aboriginal children only. English as a second language, number of notifications in the first three years, and gastroenteritis are the three child-related factors for which there is evidence of an association with the outcome for both Aboriginal and non-Aboriginal children. The mean (±SD) and median (IQR) of AEDC are substantially different for Aboriginal and non-Aboriginal children.

**Measurement and structural model fit.** Our preliminary analysis produced 17 potential predictive models. Model assessment of explanatory and predictive fit, power, utility, and simplicity reduced the potential models to a set of four competing models (Appendix III and IV in S1 File)), from which two final models were selected, one each for Aboriginal and non-Aboriginal children (Figs 2 and 3).

For the 606 Aboriginal children in the final model there was evidence for the influence of the following factors in each of the model constructs:

i. Of **pre-pregnancy** factors, Index of Relative Socio-economic Disadvantage (IRSD) was relevant and significant.

ii. Of **pregnancy** factors, gestational diabetes ($\beta$ = 0.592) and smoking during pregnancy ($\beta$ = 0.867) were both relevant and significant.

iii. Of factors **known at birth**, gender was relevant and significant ($\beta$ = 0.944), and head circumference was relevant but not significant, ($\beta$ = 0.396).

iv. Of **child-related** factors, English as a second language ($\beta$ = 0.976) was relevant and significant and preschool attendance was less relevant but significant ($\beta$ = 0.144)

Of the five domains used to form the AEDC, three domains (language and cognitive skills (school-based) ($\beta$ = 0.872), emotional maturity ($\beta$ = 0.577) and social competence ($\beta$ = 0.525)) were relevant and significant, the communication skills and general knowledge domain ($\beta$ = 0.196) was less relevant and non-significant, and the physical health and wellbeing domain was non-relevant and non-significant ($\beta$ = 0.055). The model for Aboriginal children was able to explain 22.1% of the variation in the AEDC construct and was significant (NFI = 0.96 and SRMR = 0.047).

For the 738 non-Aboriginal children in the final model there was evidence for the influence of the following factors in each of the model constructs:

i. The weighted beta coefficients of two of the **pre-pregnancy** indicators (age at pregnancy, $\beta$ = 0.475 and number of pregnancies, $\beta$ = 0.824) were relevant and significant; the other two indicators (IRSD, $\beta$ = 0.232 and occupational status of the mothers, $\beta$ = 0.227) were less relevant but significant, and the remaining indicator (educational status of the mothers, $\beta$ = 0.132) was less relevant and non-significant.

ii. Among **pregnancy** factors, maternal smoking ($\beta$ = 0.986) was relevant and significant, while having any obstetric complications was non-relevant and non-significant ($\beta$ = 0.099).

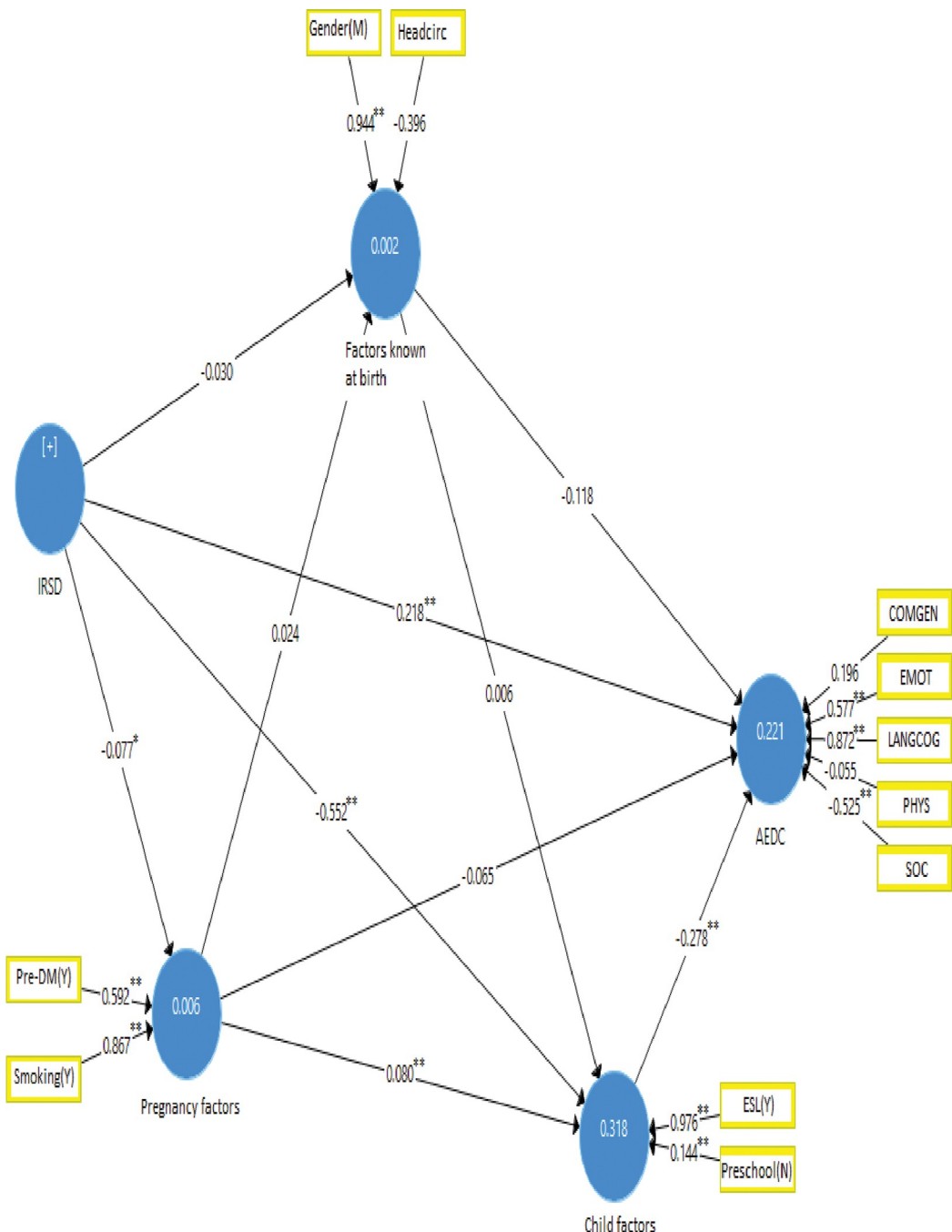

**Fig 2. Predictive model of AEDC for Aboriginal children at the first year of schooling.**

 iii. Of those factors **known at birth,** two factors (gender, β = 0.568 and gestational age, β = 0.794) were relevant but non-significant.

 iv. For those factors in the **child-related** construct, the number of notifications (β = 0.669), child protection history of the older siblings (0.397), and number of days the child stayed in hospital in the first three years (0.434) were relevant and significant.

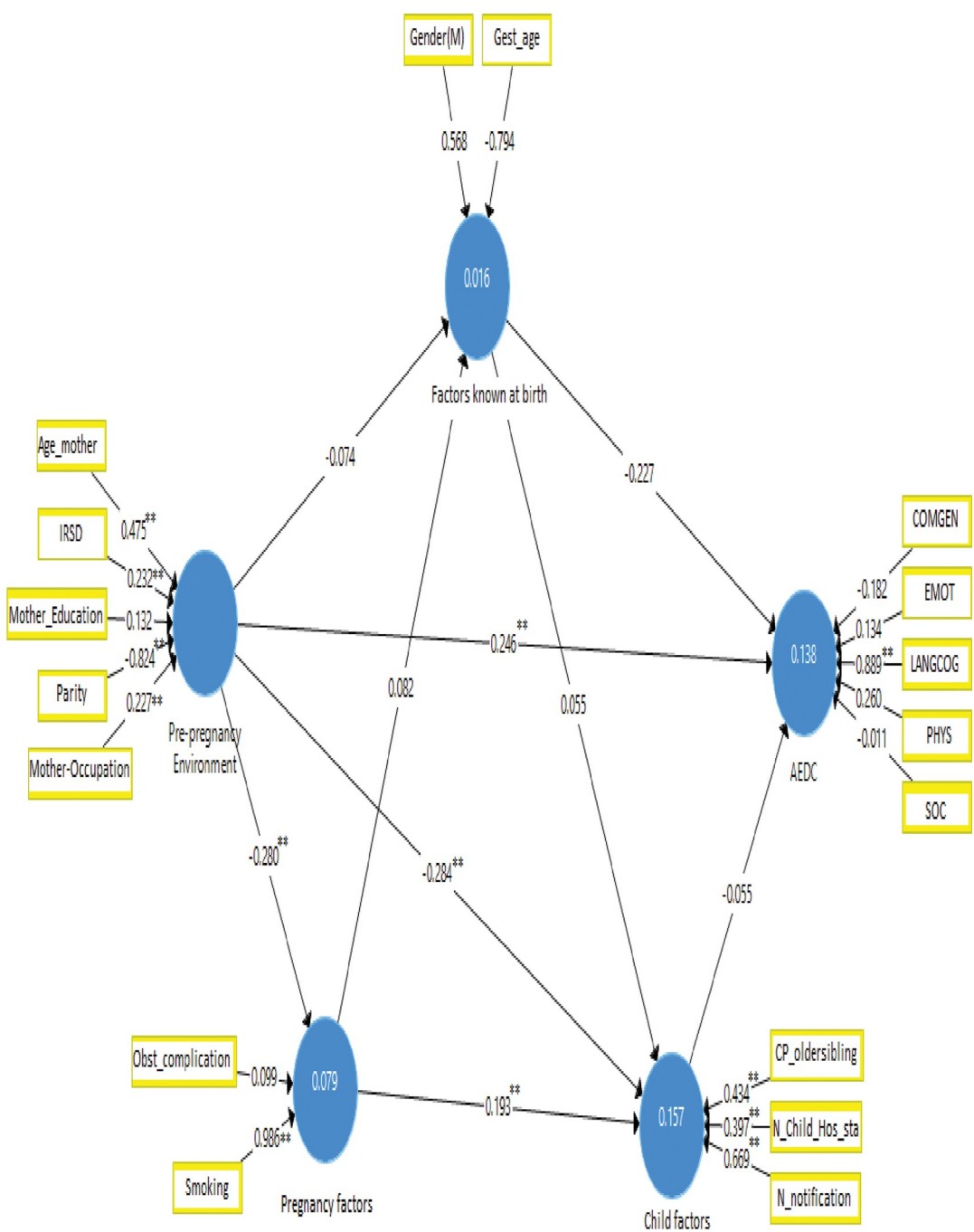

**Fig 3. Predictive model of AEDC for non-Aboriginal children at the first year of schooling.**

Of the five domains used to form the AEDC, only one (language and cognitive skills (school-based), β = 0.889) was found relevant and significant in contributing to variation, while the remaining four were less relevant and non-significant. The model for non-Aboriginal children was able to explain 13.8% of the variation accounted in the AEDC construct and was fit for explaining this variation (NFI = 0.93 and SRMR = 0.041).

## Predictive relevance of the models

We selected MAE over RMSE for evaluating the models' prediction power as prediction errors of the domains from the PLS-SEM and linear regression are not normally distributed (Appendix V in S1 File). The in-sample (with $Q^2 = 0.111$) and out-of-sample (with $Q^2_{predict} = 0.137$) prediction power of the model for Aboriginal children was higher than the acceptable threshold ($> 0$). Furthermore, most of the prediction errors of the PLS-SEM are lower than the naive linear regression (LM) model (Table 4) demonstrating the model's moderate to high out-of-sample predictive power. The direct and indirect effects of constructs having a relevance prediction are presented in Table 3. The pre-pregnancy construct (IRSD) had a direct ($\beta = 0.22$), indirect ($\beta = 0.16$) and total ($\beta = 0.38$) positive prediction effect on the AEDC outcome. The pregnancy factors construct (smoking and gestational diabetes) had an indirect ($\beta = -0.03$) and a total ($\beta = -0.09$) minor negative prediction effects on the AEDC outcome. The construct for child-related factors (English as a second language and not attending preschool) ($\beta = -0.28$) directly predicted a lower AEDC score. Further, pregnancy and child-related factors had a minor partial mediation effect on the link between IRSD and the AEDC outcome, and child-related factors fully mediated the path from pregnancy to the AEDC outcome. Overall, the model can make out-of-sample prediction with a high probability and relatively narrow prediction interval. For example, if the correct AEDC score of a child is 23, the 95% prediction interval would fall between 23±2*MAE of the AEDC construct (0.766), which is (21.468, 24.532) and arguably acceptable as the range of AEDC score runs from zero to 50.

The model for non-Aboriginal children had an in-sample (with $Q^2 = 0.052$) and out-of-sample predictive relevance (with $Q^2_{predict} = 0.068$). Furthermore, prediction errors (MAE) of all the domains of AEDC found from PLS-SEM model are less than the prediction errors found from the naive linear regression (LM) that demonstrated the model's high power for predicting the AEDC score of a child in the population (Table 4). Pre-pregnancy risk factors (older maternal age and IRSD, better occupation, and low parity) had a direct ($\beta = 0.246$) positive prediction effect on the AEDC score. In addition, there were non-significant indirect paths that increased the total prediction effect of pre-pregnancy risk factors to $\beta = 0.287$. Overall, the model can make out-of-sample prediction of AEDC scores for non-Aboriginal children with a high probability and relatively narrow prediction interval. For example, if a true observed AEDC of a child is 40, a 95% of our prediction will fall between $40 \pm 2*0.711$ (38.578, 41.422).

## Discussion

### Main findings

The study has applied a novel methodological approach to estimate how risk factors combine to predict child development (school readiness), at age five, within a life course conceptual

**Table 3. Direct, indirect and total prediction effects of predictors of AEDC score for Aboriginal and non- Aboriginal children at their first year of schooling.**

| Predictors | Direct effect (β, SE) | Indirect effect (β, SE) | Total effect (β, SE) |
|---|---|---|---|
| | | Aboriginal children | |
| Pre-pregnancy | 0.22 (0.14, 0.29) | 0.16 (0.12, 0.22) | 0.38 (0.33, 0.44) |
| Pregnancy | -0.06 (-0.14, 0.02) | -0.03 (-0.05, -0.01) | -0.09 (-0.17, -0.01) |
| Known at birth | -0.12 (-0.21, 0.04) | -0.002 (-0.025, 0.028) | -0.122 (-0.21, 0.06) |
| Child-related | -0.28 (-0.37, -0.19) | | -0.28 (-0.37, -0.19) |
| | | Non-Aboriginal children | |
| Pre-pregnancy | 0.246 (0.106, 0.349) | 0.041 (-0.002, 0.093) | 0.287 (0.160, 0.379) |
| Known at birth | -0.227 (-0.337, 0.265) | -0.003 (-0.017, 0.009) | -0.230 (-0.341, 0.270) |
| Child-related | -0.055 (-0.193, 0.048) | | -0.055 (-0.193, 0.048) |

**Table 4. Prediction power assessment of the predictive model for Aboriginal and non-Aboriginal children at their first year of schooling.**

| Domain | PLS-SEM | | LM | | PLS-SEM-LM (RMSE) | PLS-SEM-LM (MAE) |
|---|---|---|---|---|---|---|
| | RMSE | MAE | RMSE | MAE | | |
| Aboriginal children | | | | | | |
| Physical health and wellbeing | 1.784 | 1.429 | 1.785 | 1.431 | -0.001 | -0.002 |
| Social competence | 2.162 | 1.842 | 2.162 | 1.845 | 0 | -0.003 |
| Emotional maturity | 1.927 | 1.588 | 1.927 | 1.592 | 0 | -0.004 |
| Language & cognitive skills (school-based) | 2.393 | 2.014 | 2.393 | 2.012 | 0 | 0.002 |
| **Communication skills & general knowledge** | 2.913 | 2.491 | 2.913 | 2.493 | 0 | -0.002 |
| Latent AEDC | 0.932 | 0.766 | | | | |
| Non-Aboriginal children | | | | | | |
| Physical health and wellbeing | 1.375 | 1.064 | 1.381 | 1.071 | -0.006 | -0.007 |
| Social competence | 1.805 | 1.458 | 1.809 | 1.467 | -0.004 | -0.009 |
| Emotional maturity | 1.688 | 1.315 | 1.684 | 1.321 | -0.004 | -0.006 |
| Language & cognitive skills (school-based) | 1.522 | 1.107 | 1.526 | 1.110 | -0.004 | -0.003 |
| **Communication skills & general knowledge** | 2.121 | 1.723 | 2.126 | 1.723 | -0.005 | 0 |
| Latent AEDC | 0.969 | 0.711 | | | | |

**Note**: PLS-SEM: Partial least square structural equation modeling; LM: Linear model; RMSE: Root mean square error; MAE: Mean absolute error.

framework. Our study demonstrates different predictive models for Aboriginal and non-Aboriginal children. The models are fit for explanation and for making out-of-sample prediction. The difference between predictive models for Aboriginal and non-Aboriginal children speaks to the need for a nuanced response that addresses the separate influences for the two groups of children. A limitation for the model is that the AEDC construct for measuring child development (school readiness) was not balanced across all domains. For Aboriginal children the variation in AEDC outcomes between children was influenced by measurement in only three domains–emotional maturity, social competence, and language and cognitive skills (school-based) while for non-Aboriginal children only the single domain of language and cognitive skills (school-based) was influential in variation in AEDC measurement between children.

Our study demonstrated that, for Aboriginal children, the prediction of school readiness was influenced by a wider range of clusters, including maternal and pre-pregnancy factors (community-level socio-demographic disadvantage), child-related factors (English as second language and not attending preschool), and pregnancy factors (gestational diabetes and maternal smoking). Community-level sociodemographic disadvantage explained 51%, child-related factors explained 37% and pregnancy factors explained 12% of the influential paths to the AEDC outcome. For non-Aboriginal children, the prediction of child development (school readiness) is dominated by the direct effects of maternal and pre-pregnancy factors, with teenage pregnancy, increased parity, community-level sociodemographic disadvantage and lower-level of occupation of the primary carer predicting increased vulnerability and lower school readiness. Parity contributed 47%, teenage pregnancy 27%, community-level sociodemographic disadvantage 13.2% and occupation 13% of the variation within a cluster of pre-pregnancy factors. The models also demonstrated the interplay of these factors in predicting child development (school readiness). For example, for Aboriginal children, pregnancy and child-related factors partially mediated the link between pre-pregnancy and AEDC outcomes (explaining 42% of the total variation in pre-pregnancy factors); and child-related factors fully mediate the link between pregnancy factors and AEDC outcomes. Most of these risk factors

have been previously reported to explain child language and cognitive development in first year of formal schooling [15]. The identified predictors of child development (school readiness) in our study are also consistent with a previous explanatory study in NT [54] and predictive studies conducted in other states of Australia using a dichotomous AEDC outcome measure for AEDC [24,59,85].

Despite the limitation of the modelled outcome not being explained equally by all AEDC domains, this study provides insight into areas in which efforts should be made to improve child development (school readiness). Most importantly, the predictive model highlights the cluster of factors in which there should be focus for improving a model for early identification of children at higher risk of development and school readiness. The model also informs how risk factors appearing at different times in the life course are interrelated in their influence on development. Our analysis is consistent with recent recommendation promoting a life course approach (from conception to the first three years) to childhood development [86].

The generalizability and validity of the predictive models are subject to the following limitations. First, the poor fit of the formatively measured domains of the AEDC for both Aboriginal and non-Aboriginal population limits the potential prediction of an AEDC construct that encompasses five developmental domains. A previous validation study reported the Australian version of the Early Development Instrument (AvEDI) as valid [66]. However, this validation study specified the five AEDC domains as reflective measures although they theoretically appear to be distinct concepts that should be constructed as formative [78,79,87]. Our study fitted the five domains as formative measures of AEDC and found that only three domains for Aboriginal children (emotional maturity, language and cognitive skills (school-based), and social competence) and one domain for non-Aboriginal children (language and cognitive skills (school-based)) were relevant and significant. Further investigation is required to determine if the poor fit of the AEDC measure is a consequence of the ability of the measure to discriminate varying levels of performance between children or whether some children were inappropriately assessed during data collection. Reliable data collection requires an ongoing investment in training for teachers and in the provision of classroom assistance (including Aboriginal education support) during collection. Second, our assumption that the predictors measured for children at age five were the same at the time the mother was pregnant may not be true, which affects temporal validity. However, the use of this theoretical framework of child development (school readiness) to conceptualize the causal relationship of these predictors is legitimate [70]. Third, the much higher level of missing data for individual-level variables for NT Aboriginal children (for example maternal education missing for 41% of Aboriginal children compared with 20% of non-Aboriginal children) may have introduced a level of bias into the estimates. A fourth limitation is that the homogeneity of some covariates across the population may reduce the discriminatory ability of individual-level variables for Aboriginal children. As an example, among children with maternal education information, 63% of Aboriginal children have maternal education level recorded as not completing school, compared to 25% of non-Aboriginal children. As such, our analysis might understate a gradient in the association between individual-level markers of socio-economic disadvantage and school readiness for Aboriginal children. A previous study of risks and protective factors for school readiness among Australian Aboriginal children reported the importance of parent education and emphasized the benefit of maternal high school completion [88]. A fifth limitation is that potentially important covariates such as maternal mental health (pre and postnatal period), exposure to family violence, child parenting practice, maternal nutrition during pregnancy and childhood nutritional status were not available and should be considered for future studies.

## Policy implications

Our findings have several policy implications. Our study demonstrated the high prevalence of socio-economic disadvantage experienced by NT Aboriginal children from pre-natal period and extending through early childhood largely occurred at the community-level, suggesting the importance of 'place-based' strategies such as an integrated services model for early childhood services and Aboriginal community and health services [89]. The demonstrated benefit of preschool attendance and negative influence of a non-English background for school readiness, for Aboriginal children, speaks to the importance of preschool participation and English language support at school entry in remote communities.

## Place-based strategies

Research, both international and in Australia, has identified that integrated services are more effective in meeting the needs of communities and families whose personal resources are continually compromised by more urgent, complex and multiple needs. An example of discontinuity between services experienced by many families is the transition between preschool and school. For many Aboriginal children there is a cultural and linguistic gap between home and school or home and other service providers. Integrated or seamless preschool and early childhood care and education programs could be a way to provide early experiences of 'schooling culture' and support successful transitions to Western-style school. In Australia, several long-established not-for-profit early childhood service providers have developed integrated models to meet their clients' needs more effectively. For example, in Western Australia, Ngala provides a range of universal, targeted and intensive services, including early childhood education and care, parenting groups and overnight support for mothers and babies [90]. Other examples of long-established organizations developing integrated programs in Australia includes Best chance Child and Family Care in Victoria [91], The Infants' Home in New South Wales [91], Gowrie in South Australia [92], the Doveton high-quality education and wrap-around community-service model in Victoria [93], and Tasmania's child and family centres [94]. Though these place-based learning centres had a long history, their effectiveness has not been comprehensively evaluated with the exceptions of Tasmania's child and family centre and the Doveton community service model where recent studies have reported the learning centres as a welcoming place to develop positive child and family outcomes [95–97]. A Productivity Commission report in 2020 commented on the continuing fragmentation of early childhood services in the NT resulting in ongoing gaps and duplication of funding to services which often did not address community interests or needs [89]. Future directions should be guided by studies that unpack the school-related or community factors that enable improved educational outcomes [55].

## Preschool and provision of English language support

Our study also demonstrates that two immediately available strategies for improving school readiness and educational outcome are preschool and the provision of English language support for Aboriginal children throughout schooling, including late exit bilingual models where feasible. This finding supports the inclusion of the new Closing the Gap targets relating to Aboriginal educational outcomes to increase the enrolment and participation in early childhood education to 95% by 2025 [51]. A review of interventions designed to improve school readiness of Aboriginal children revealed that "*high quality evidence is sparse and effective interventions are few in number*". While the importance of preschool is widely acknowledged, a previous NT study indicated that the initial benefits of preschool can 'fade out' unless they are reinforced by regular attendance and effective engagement with school learning in the early years of primary school [88]. Silburn et al highlighted the necessity of policy and services

supporting children's transition into formal school learning extending through to at least Year 3 [55].

## Conclusion

This predictive model for child development (school readiness) identified separate models for Aboriginal and non-Aboriginal children in the NT. For Aboriginal children the most influential predictors are community-level sociodemographic disadvantage and preschool attendance both of which point to structural inequalities within policy and program design. The most influential predictors for non-Aboriginal children are maternal parity, maternal age, community level socio-demographic disadvantage, and maternal occupation. The models give insight into the interplay of these factors at different stages of a child's development and informs service and policy responses such as language diversity and cultural inclusion to improve school education outcomes. However, the measurement of AEDC domains and educator's assessment practice needs to be reviewed to ensure balanced discernment of all five domains of measurement.

## Supporting information

**S1 File. Predictive model.**
(DOCX)

## Acknowledgments

The authors would like to acknowledge the support by the NT Government Departments of Health; Education; Territory Families, Housing and Communities; Attorney General and Justice; Chief Minister and Cabinet; Treasury and Finance; and Police, Fire and Emergency Services, through the Child and Youth Development Research Partnership (CYDRP). We also acknowledge the use of data from the Australian Early Development Census (AEDC). The AEDC is funded by the Australian Government Department of Education, Skills and Employment. We thank SA NT DataLink personnel for their technical and administrative assistance in the linkage of datasets and all data custodians for their support with retrieval, preparation and release of the research datasets. The views expressed in this publication are those of the authors and should not be attributed to the government departments who have supplied the data for the study.

## Author Contributions

**Conceptualization:** Vincent He, Jiunn-Yih Su, Steven Guthridge.

**Data curation:** Abel Fekadu Dadi, Vincent He.

**Formal analysis:** Abel Fekadu Dadi.

**Investigation:** Abel Fekadu Dadi, Vincent He, Steven Guthridge.

**Methodology:** Abel Fekadu Dadi, Vincent He, Steven Guthridge.

**Project administration:** Steven Guthridge.

**Resources:** Steven Guthridge.

**Supervision:** Steven Guthridge.

**Validation:** Abel Fekadu Dadi, Vincent He, Georgina Nutton, Jiunn-Yih Su, Steven Guthridge.

**Visualization:** Abel Fekadu Dadi.

**Writing – original draft:** Abel Fekadu Dadi.

**Writing – review & editing:** Abel Fekadu Dadi, Vincent He, Georgina Nutton, Jiunn-Yih Su, Steven Guthridge.

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
