## [Decision Letter · Decision Letter 0]

11 Sep 2022

PONE-D-22-21307Predicting child development and school readiness, at age 5, for Aboriginal and non-Aboriginal children in Australia’s Northern TerritoryPLOS ONE

Dear Dr. Fekadu,

Thank you for submitting your manuscript to PLOS ONE. After careful consideration, we feel that it has merit but does not fully meet PLOS ONE’s publication criteria as it currently stands. Therefore, we invite you to submit a revised version of the manuscript that addresses the points raised during the review process. As you can see below, the reviews are mixed, with R1 being the more positive of the two; though they still point to a number of areas in need of attention. R2 calls for the voices of First Nations people to be integrated more completely in the manuscript and for the need to address assumptions that Indigenous children will be forced into a non-Indigenous school system. Other points raised and as these are clear I will not repeat them here. But needless to say, all need to be addressed.

We look forward to receiving your revised manuscript.

Kind regards,

Mark Nielsen, Ph.D.

Academic Editor

PLOS ONE

Journal Requirements:

"The study was supported by a grant from the Northern Territory Government through the Child and Youth Development Research Partnership. The funding body was not involved in the study design, data analysis, interpretation of data, or preparation and publication of research papers."

"We acknowledge the seven NT Government agencies who participate in the Child Youth and Development Research Partnership (CYDRP) for their sustained support."

"The study was supported by a grant from the Northern Territory Government through the Child and Youth Development Research Partnership. The funding body was not involved in the study design, data analysis, interpretation of data, or preparation and publication of research papers."

Reviewers' comments:

Reviewer's Responses to Questions

**Comments to the Author**

1. Is the manuscript technically sound, and do the data support the conclusions?

Reviewer #1: Yes

Reviewer #2: Partly

2. Has the statistical analysis been performed appropriately and rigorously? 

Reviewer #1: Yes

Reviewer #2: I Don't Know

3. Have the authors made all data underlying the findings in their manuscript fully available?

Reviewer #1: No

Reviewer #2: No

4. Is the manuscript presented in an intelligible fashion and written in standard English?

Reviewer #1: Yes

Reviewer #2: Yes

5. Review Comments to the Author

Reviewer #1: This is a generally well written paper. I have a few comments that need to be addressed:

1. In your Discussion, can you clarify what is novel in your findings? On page 26 you state that much of what you found has been found before - so can you elaborate on what is truly new?

2. Abstract - in your Conclusion, which aspects of a child's development need to be captured equally - ie are there dimensions missing which should be highlighted in the abstract?

3. On page 5, for ref 50, can you add in the year of the finding that only 41% of Aboriginal children attended pre-school?

4. Design and Population - when you say you used unit-record level data, does unit= the child? Would be clearer to say this

5. Covariates- in your Discussion, can you say what key covariates you think are missing from the available datasets that should be captured in any future datasets? eg family violence? parenting? quality of preschool? other key determinants of child development?

6. Data analysis - for your logistic regression for non-Aboriginal children, how did you decide on your cut point (I may have missed this)?

7. By using CCA approach, you lost a number of children - can you say how those included vs dropped differed? Does this have any implications for the generalisability of your findings?

8. On page 29, you discuss the need for "integrated or seamless preschool and early childhood care and education programs" ...easy to say but hard to do - can you give any examples of successful programs?

Reviewer #2: Thank you for the opportunity to read this manuscript. Clearly a lot of work went into it and tackles an important issue – Aboriginal child development, an area of great importance to Aboriginal communities. Large sections of the paper are very well-written.

However, I have several major concerns. I struggled to hear the voices of Aboriginal communities and researchers in this work (apologies to any of the authors who are Aboriginal and/or Torres Strait Islander). In recent years, a number of eminent Aboriginal researchers have tried to guide research in a new direction and I think this manuscript would be strengthened by following some of the recommendations of those researchers (e.g. Maggie Walter, Stephen Harfield’s paper Assessing the quality of health research from an Indigenous perspective: the Aboriginal and Torres Strait Islander quality appraisal tool). In particular, this paper would benefit from being written from a strengths-based perspective. Also, Aboriginal should always have a capital ‘A’ and this needs to be corrected in the supplementary material. Similarly, the noun “Aboriginals” could be replaced in the supplementary material with something like “Aboriginal people”.

I felt there was sometimes a blurring of the distinct concepts of child development, readiness for a school system that was not set up for Aboriginal children, and AEDC scores. I think the manuscript could benefit from being careful to articulate which concept is being referred to throughout.

Could the authors provide more detail about why the AEDC scores were measured formatively – I would have thought that the construct (child development) informs the indicators (AEDC scores), rather than the other way around, so a more detailed explanation or diagram would help (I note that Fig 1 has been included, but I assume that the correct reading of Fig 1 is that ‘true’ physical health and well-being informs ‘true’ child development and the AEDC score for physical health is not in this figure, but if it was, there would be an arrow from ‘true’ physical health and well-being to AEDC score).

Underlying the discussion appears to be an assumption that Aboriginal children should be prepared to a school system set up largely by non-Aboriginal people. I note that many Aboriginal families would like the system to fit to their children’s needs rather than the other way round.

The results from the complete case analysis (CCA), indicator variable for missing data approach and MICE (Supplementary Table 1) were very different – some variables were significant CCA but not the other approaches and some coefficients changed direction. Given this, how would their conclusions have changed if MICE was the preferred approach?

I am curious about the finding that missingness did not depend on the outcome. Looking at Table 1, missingness was much more common among Aboriginal children and AEDC scores were lower for Aboriginal children, so I would have expected an association between missingness and AEDC scores. It would be interesting to see data on AEDC scores for children missing data vs AEDC scores for children not missing data. How did the authors conclude missingness did not depend on the outcome?

Some of the data analysis section is hard to follow and could be rewritten and restructured so it is clearer what the aim of each step is and so that information flows in a logical order (e.g. “in-sample” appears at line 254, but “in-sample” and “out-of-sample” is not explained until later.

The decision-making about how the final models were selected is difficult to follow (e.g. line 324 – it is not clear how the authors ended up with 17 competing models and how they went from those 17 to 4). There is mention of the supplementary material, but there appear to only be 15 models in the supplementary material and it’s not clear what the 4 final models are, nor how the 2 presented in the main paper were chosen. Perhaps a diagram could explain this.

Minor suggestions/comments:

Line 121: Suggest rewriting the first sentence as not currently clear. However, the rest of this paragraph is very good and perhaps the ideas in this paragraph could inform the rest of the paper a bit more.

Line 182: Typo

Line 200: Specify the variables more exactly (e.g. what were the categories of education and employment?)

Line 223: Explain what total AEDC summary scale scores are

Line 229 and 230: What are these models? What is the binary outcome for the non-Aboriginal children?

Line 238: Include how many children were excluded and on what grounds.

Line 323: What does “Added percentile of AEDC domains” mean? Need to clarify that p-values at the end of Table 1 are for comparisons of Aboriginal and non-Aboriginal children while p-values for the rest of the table for comparisons of AEDC values and the variable.

6. PLOS authors have the option to publish the peer review history of their article (what does this mean?). If published, this will include your full peer review and any attached files.

Reviewer #1: **Yes: **Harriet Hiscock

Reviewer #2: No

---

## [Decision Letter · Decision Letter 1]

26 Apr 2023

PONE-D-22-21307R1Predicting child development and school readiness, at age 5, for Aboriginal and non-Aboriginal children in Australia’s Northern TerritoryPLOS ONE

Dear Dr. Dadi,

Thank you for submitting your manuscript to PLOS ONE. After careful consideration, we feel that it has merit but does not fully meet PLOS ONE’s publication criteria as it currently stands. Therefore, we invite you to submit a revised version of the manuscript that addresses the points raised during the review process.

We look forward to receiving your revised manuscript.

Kind regards,

Gursimran Dhamrait, Ph.D

Academic Editor

PLOS ONE

Reviewers' comments:

Reviewer's Responses to Questions

**Comments to the Author**

1. If the authors have adequately addressed your comments raised in a previous round of review and you feel that this manuscript is now acceptable for publication, you may indicate that here to bypass the “Comments to the Author” section, enter your conflict of interest statement in the “Confidential to Editor” section, and submit your "Accept" recommendation.

Reviewer #2: (No Response)

Reviewer #3: (No Response)

2. Is the manuscript technically sound, and do the data support the conclusions?

Reviewer #2: Partly

Reviewer #3: Partly

3. Has the statistical analysis been performed appropriately and rigorously? 

Reviewer #2: I Don't Know

Reviewer #3: I Don't Know

4. Have the authors made all data underlying the findings in their manuscript fully available?

Reviewer #2: No

Reviewer #3: Yes

5. Is the manuscript presented in an intelligible fashion and written in standard English?

Reviewer #2: Yes

Reviewer #3: Yes

6. Review Comments to the Author

Reviewer #2: I don't believe the authors have adequately addressed my comment re missing data, or Reviewer 1's comment, and in the process of re-reading it, I realised the authors may not have accurately described the recommendations laid out in the two papers on missing data that they referenced.

It was not clear from what they wrote whether the authors thought their data were missing at random (MAR) or missing not at random (MNAR). They wrote “Missingness in the covariates was related to the outcome variable. Children who missed observation on these variables were more likely to be developmentally vulnerable in one or two domains. So, missingness was not random and Multiple Imputation would introduce bias. CCA is the appropriate method.” Having missingness in the covariates related to the outcome variable, does not (of itself) distinguish between MAR (which, despite its name, is not completely random) and MNAR and there is no 'test' to distinguish between MAR and MNAR. Given that it is hard to know whether data is MAR or MNAR, it is also not always clear whether CCA or MI is preferable and often both are used and the results compared.

Despite this, the authors claim that CCA is preferable, “In this case, methodological studies have reported that multiple imputation (MI) tends to be biased away from the null and complete case analysis (CCA) is a better approach (83, 84)”, I note that reference 84 actually starts the Discussion section with “Our comparison of MAR-based MI with CC analysis for regression models with one or two incomplete covariates shows no universally consistent advantage of one method over the other, although MI appears to be superior across a wider range of settings.“ For reference 83, despite the title “Accounting for missing data in statistical analyses: multiple imputation is not always the answer”, the authors themselves chose MI for the analysis they were interested in.

Given all of this, the approach of doing both CCA and MI in this study was welcome, but the results from both approaches should be discussed. Re-reading my comment, I realise I could have made this point more clearly. The case is better made in Sterne JA, White IR, Carlin JB, Spratt M, Royston P, Kenward MG, Wood AM, Carpenter JR. Multiple imputation for missing data in epidemiological and clinical research: potential and pitfalls. Bmj.

“Where complete cases and multiple imputation analyses give different results, the analyst should attempt to understand why, and this should be reported in publications.”

The sentence “In this case, methodological studies have reported that multiple imputation (MI) tends to be biased away from the null and complete case analysis (CCA) is a better approach (83, 84)” needs to be re-written to better reflect the content of the papers they are referencing.

Reviewer #3: Given this paper has already been reviewed by others, I have focused my attention on the response of the authors to the original reviewer comments—although I have a few additional (minor) comments on other aspects of the paper. Overall, I thought the paper was well written, included a thorough approach to the analysis and is centred on an important topic in research, policy and practice.

My main area of concern was highlighted by Reviewer 1 (point 6): and concerned the approach to dealing with data missing not at random. The authors response suggests that a CCA is preferred to a multiple imputation approach. This may be the case but I am not an expert in these methodological issues – my readings suggest, on balance, that this is a fair argument. However, it remains the fact that around 52-60% of the Aboriginal and non-Aboriginal sample were included in the final prediction models. Given this scale of missingness and likelihood of bias, this appears to me to be a major issue in terms of the impact on the selection of variables and estimated effects. The sensitivity analysis, which highlights differences in the results between the CCA and MICE analyses, doesn’t appear to ease this concern.

Perhaps part of the issue here is the (from what I can tell) reliance on Model I results in informing the selection of variables in the prediction models – there appears to be a significantly smaller sample for the CCA analysis compared with Models II and III.

In any case, all of this, of course, places limitations on the implications of the final model results for policy and practice—and while the issues of bias are mentioned as a limitation, I’m not sure the authors do enough to discuss this concern or asses the likelihood of it not being serious.

Additional comments

- The terms ‘positive’ and ‘poorer’ outcomes in the Abstract should be defined or a more specific term used. It also wasn’t clear to me what was meant by the term ‘not equally captured’

- I would have preferred the Conclusion of the Abstract to tell us what factors are important, based on the model results

- First para of page 5: assume this relates to all Australian children (not just NT)?

- Page 15, line 319: should clarify that the univariable association is with the AEDC total score (although the effect is not reported; instead it reflects that there was a significant association); also, from what I can tell, these are associations in the regression models of step 1 as opposed to the predictive model? If I have interpreted this correctly then a few tweaks need to be made to the text and the table titles that follow

- Page 30: the inclusion of more details on place-based strategies is very useful. Is there any information on whether these have been evaluated and therefore effective?

- Page 31 (Conclusion): I don’t disagree that SES and preschool attendance are implicated in structural inequalities, however, I’m not sure that the presence of these factors in the model for Aboriginal children suggests this given it is a within-Aboriginal sample model.

7. PLOS authors have the option to publish the peer review history of their article (what does this mean?). If published, this will include your full peer review and any attached files.

Reviewer #2: No

Reviewer #3: No

---

## [Author Response · Author response to Decision Letter 1]

11 May 2023

Response to reviewers' comment has been attached to submission folder.

---

## [Decision Letter · Decision Letter 2]

21 Nov 2023

PONE-D-22-21307R2Predicting child development and school readiness, at age 5, for Aboriginal and non-Aboriginal children in Australia’s Northern TerritoryPLOS ONE

Dear Dr. Dadi,

Thank you for submitting your manuscript to PLOS ONE. I apologise it has taken so long to respond to you which has been largely due to our endeavours to find Indigenous researchers with an interest in this topic willing to review this paper. After careful consideration, we feel that it has merit but does not fully meet PLOS ONE’s publication criteria as it currently stands. Therefore, we invite you to submit a revised version of the manuscript that addresses the points raised during the review process particularly around how you have handled the missing data.

We look forward to receiving your revised manuscript.

Kind regards,

Caroline Watts, PhD

Academic Editor

PLOS ONE

Journal Requirements:

Reviewers' comments:

Reviewer's Responses to Questions

**Comments to the Author**

1. If the authors have adequately addressed your comments raised in a previous round of review and you feel that this manuscript is now acceptable for publication, you may indicate that here to bypass the “Comments to the Author” section, enter your conflict of interest statement in the “Confidential to Editor” section, and submit your "Accept" recommendation.

Reviewer #2: (No Response)

2. Is the manuscript technically sound, and do the data support the conclusions?

Reviewer #2: Partly

3. Has the statistical analysis been performed appropriately and rigorously? 

Reviewer #2: N/A

4. Have the authors made all data underlying the findings in their manuscript fully available?

Reviewer #2: No

5. Is the manuscript presented in an intelligible fashion and written in standard English?

Reviewer #2: Yes

6. Review Comments to the Author

Reviewer #2: The authors have undertaken more work examining the missingness in the data. This version states, “Furthermore, we also checked that the missingness was not dependent on the outcome (AEDC) after adjusting for the other covariates”. It is not clear how this was checked and there has a confusing history about the nature of the missing data.

In the first draft, the authors stated that missingness was not dependent on the outcome. When this was queried, based on results in Table 1, the authors advised that this was an error and revised the text to, “missingness was associated with the outcome (AEDC) i.e., children who missed observation on these variables were more likely to developmentally vulnerable on one or two domains”. In this version, they state that missingness is not dependent on the outcome after adjusting for the other covariates.

Unfortunately, the authors did not provide evidence to support this new statement, nor a description of the methodology used. If there is no association between missingness and the outcome, conditional on the other covariates, I accept that the references used are appropriate. If not, my recommendation would remain that:

- No new analysis is required. Only some revision of the written material is required.

- Multiple approaches are appropriate

- Differences in results between the complete case analysis and the 2 sensitivity analyses should be discussed

- The wording around those references should be changed.

7. PLOS authors have the option to publish the peer review history of their article (what does this mean?). If published, this will include your full peer review and any attached files.

Reviewer #2: No

---

## [Author Response · Author response to Decision Letter 2]

3 Dec 2023

I have uploaded a rebuttal letter with the submission.

---

## [Editor Report · Decision Letter 3]

6 Dec 2023

Predicting child development and school readiness, at age 5, for Aboriginal and non-Aboriginal children in Australia’s Northern Territory

PONE-D-22-21307R3

Dear Dr. Dadi,

We’re pleased to inform you that your manuscript has been judged scientifically suitable for publication and will be formally accepted for publication once it meets all outstanding technical requirements.

Kind regards,

Caroline Watts, PhD

Academic Editor

PLOS ONE

---

## [Editor Report · Acceptance letter]

11 Dec 2023

PONE-D-22-21307R3 

Predicting child development and school readiness, at age 5, for Aboriginal and non-Aboriginal children in Australia’s Northern Territory 

Dear Dr. Dadi:

I'm pleased to inform you that your manuscript has been deemed suitable for publication in PLOS ONE. Congratulations! Your manuscript is now with our production department. 

Kind regards, 

on behalf of

Dr. Caroline Watts 

Academic Editor

PLOS ONE